# The Relationship Between Dietary Patterns, Cognition, and Cardiometabolic Health in Healthy, Older Adults

**DOI:** 10.3390/nu16223890

**Published:** 2024-11-14

**Authors:** Felicity M. Simpson, Alexandra Wade, Ty Stanford, Maddison L. Mellow, Clare E. Collins, Karen J. Murphy, Hannah A. D. Keage, Montana Hunter, Nicholas Ware, Daniel Barker, Ashleigh E. Smith, Frini Karayanidis

**Affiliations:** 1Functional Neuroimaging Laboratory, School of Psychological Sciences, University of Newcastle, Callaghan, NSW 2308, Australia; 2Healthy Minds Research Program, Hunter Medical Research Institute, Newcastle, NSW 2305, Australia; daniel.barker@hmri.org.au; 3Alliance for Research in Exercise, Nutrition and Activity (ARENA), Allied Health and Human Performance, University of South Australia, Adelaide, SA 5001, Australiaty.stanford@unisa.edu.au (T.S.); maddison.mellow@unisa.edu.au (M.L.M.); ashleigh.smith@unisa.edu.au (A.E.S.); 4College of Education, Psychology and Social Work, Flinders University, Bedford Park, SA 5042, Australia; 5School of Health Sciences, College of Health, Medicine and Wellbeing, The University of Newcastle, Callaghan, NSW 2308, Australia; clare.collins@newcastle.edu.au; 6Food and Nutrition Research Program, Hunter Medical Research Institute, New Lambton Heights, NSW 2305, Australia; 7Clinical & Health Sciences, University of South Australia, Adelaide, SA 5001, Australia; 8Justice and Society, University of South Australia, Adelaide, SA 5001, Australia; hannah.keage@unisa.edu.au; 9School of Psychology and Vision Sciences, University of Leicester, Leicester LE1 7RH, UK; mrh45@leicester.ac.uk

**Keywords:** dietary patterns, cardiometabolic health, healthy cognitive ageing, dementia prevention

## Abstract

Background: Healthy dietary patterns can support the maintenance of cognition and brain health in older age and are negatively associated with cardiometabolic risk. Cardiometabolic risk factors are similarly important for cognition and may play an important role in linking diet to cognition. Aim: This study aimed to explore the relationship between dietary patterns and cognition and to determine whether cardiometabolic health markers moderate these relationships in older adulthood. Design: A cross-sectional analysis of observational data from the baseline of the ACTIVate study. Participants: The cohort included 426 cognitively normal adults aged 60–70 years. Methods: The Australian Eating Survey (AES) Food Frequency Questionnaire was used to collect data on usual dietary intake, along with additional questions assessing intake of dietary oils. Principal component analysis (PCA) was applied to reduce the dimensionality of dietary data. Cardiometabolic risk was quantified using the metabolic syndrome severity score (MetSSS). Tests from the Cambridge Neuropsychological Test Automated Battery (CANTAB) were used to derive composite scores on four cognitive domains: processing speed, executive function, short-term memory, and long-term memory. Results: Three dietary patterns were identified using PCA: a plant-dominant diet, a Western-style diet, and a meat-dominant diet. After controlling for age, sex, total years of education, energy intake, and moderate-to-vigorous physical activity (MVPA), there was a small, negative association between the meat-dominant diets and long-term memory. Subsequent moderation analysis indicated that MetSSS significantly moderated this relationship. Conclusions: Findings highlight the link between diet, cardiometabolic health, and cognitive function in older, cognitively healthy adults. However, longitudinal studies are needed to confirm observations and evaluate the dynamics of diet, cardiometabolic health, and cognitive function over time.

## 1. Introduction

The number of people living with dementia is projected to reach 152 million worldwide by 2050 [1]. In the absence of effective dementia treatments, research has focused on addressing modifiable lifestyle factors that contribute up to 45% of dementia cases, in people aged 65 years and older [2]. While a healthy diet is linked to better cognitive performance, it is important to understand how this relationship is affected by the presence of other modifiable factors, such as cardiometabolic health, which is also known to deteriorate as we age.

Nutritional epidemiology has historically focused on the impact of isolated nutrients and their diet–disease relationships [3]. Studies on isolated nutrients overlook complex dietary interactions between food groups [4]. As a result, randomised controlled trials (RCTs) using nutritional supplements to enhance cognition typically show poor translation of results [5]. Since specific foods or nutrients are rarely consumed in isolation, examining dietary patterns allows for the interplay between nutrition and cognitive outcomes to be explored, capturing the synergistic effects of various nutrients and components of the food matrix [6]. Additionally, dietary patterns are likely easier to communicate to the public, potentially facilitating the uptake of recommendations. Moreover, there is evidence that the characterisation of overall dietary patterns yields more consistent and robust associations with cognitive health compared to isolated nutrients, underscoring the importance of this approach [7].

The Mediterranean diet (MedDiet) is characterised by high consumption of plants, whole grains, legumes, nuts, and fish, moderate portions of poultry, dairy foods, and red wine, extra virgin olive oil as the main culinary fat, and a limited intake of red meats and discretionary foods [8,9]. High alignment with a MedDiet has been associated with better cognitive health and a reduced risk of dementia in epidemiological studies (see reviews [10,11]. These findings are supported by experimental evidence from studies such as PREDIMED, one of the largest diet interventions [12] which explicitly recruited participants with high cardiovascular disease risk. After five years, participants exposed to the MedDiet had less cognitive decline than those in the low-fat diet control group. MedDiet adherence has also been associated with larger brain volume and cortical thickness lower total brain volume atrophy, improved structural connectivity, as well as a lower burden of white matter hyperintensities and cerebral infarcts [13,14,15,16,17,18,19].

In contrast, a Western-style dietary pattern is characterised by high consumption of processed foods, refined carbohydrates, saturated fat, and sodium, with low consumption of fruit and vegetables. In older adults, a Western-style diet is consistently associated with poor health and cognitive outcomes (see review [20]). Longitudinal studies have shown that a Western-style dietary pattern is associated with a greater decline in global cognitive abilities [21,22,23]. The Western-style diet is also associated with poorer performance in vocabulary and phonemic fluency, visuospatial functioning, memory, and executive function in men [24,25,26,27]. Greater alignment with a Western-style diet is also associated with poorer markers of brain health, including lower hippocampal volume in older adults [28] and impaired hippocampal function and integrity [29]. This is notable, given that the hippocampus is a brain region crucial in memory formation, and atrophy is a hallmark of dementia [30]. Therefore, Western-style diets may be linked to Alzheimer’s disease pathology [31].

Despite substantial evidence linking diet to cognitive outcomes, the interaction between dietary patterns and other factors that influence cognitive health, particularly cardiometabolic risk factors, remains poorly understood. Dietary patterns can affect the presence of cardiometabolic risk factors. Adherence to healthful dietary patterns (i.e., whole-food, nutrient-dense) is associated with lower cardiometabolic risk (see reviews [32,33,34,35]). Conversely, consuming a diet characterised by poor nutrient quality, such as the Western-style diet, is associated with adverse cardiometabolic outcomes, including hypertension, obesity, dyslipidaemia, and impaired glucose metabolism [34].

The prevalence of these cardiometabolic risk factors markedly increases in mid-late life [36] and is strongly associated with cognitive functioning. Individuals with a higher risk factor burden during midlife (e.g., hypertension, obesity, and dyslipidaemia) are more likely to have or develop cognitive impairment and dementia [37,38,39]. Despite this, most studies that examine the association between diet and cognition in older adults focus primarily on healthful diets, such as plant-based diets, the Dietary Approaches to Stop Hypertension (DASH), MediMind (combination of DASH and MedDiet) and the MedDiet (see reviews [10,40,41,42,43]). As dietary patterns are inherently complex and multidimensional, it is essential to consider the broad spectrum of dietary patterns to fully understand their relationships with cognitive and cardiometabolic health.

The current study aimed to explore relationships between dietary patterns and cognition in a healthy, community-dwelling group of older adults. We hypothesised that at least two dietary patterns would emerge using principal component analysis (PCA): a healthful diet and a suboptimal diet; we anticipated that the healthful diet would be associated with better performance in cognitive domains, whereas the poor-quality diet would be associated with poorer cognitive performance. A second aim was to evaluate whether the relationship between dietary patterns and cognitive performance is moderated by cardiometabolic health. To do this, we tested how a metabolic syndrome severity score [44] moderated relationships between diet and cognition. Understanding the potential moderation effect of cardiometabolic health on the diet–cognition relationship can provide valuable insights into potential avenues for personalised preventive strategies.

## 2. Materials and Methods

### 2.1. Study Design

The present study used data from the baseline phase of the ACTIVate cohort study [45]. The ACTIVate study was conducted in accordance with the Declaration of Helsinki, following ethics approval obtained from the University of South Australia and the University of Newcastle Human Research Ethics Committee (202639). The study was registered with the Australian New Zealand Clinical Trials Registry (registration number ACTRN12619001659190) on 27 November 2019.

### 2.2. Participants

The ACTIVate cohort consists of 426 healthy, community-dwelling adults recruited at the University of South Australia (n = 226) and the University of Newcastle (n = 200). Participants were eligible for inclusion if they were aged between 60 and 70 years, fluent in English, and did not have a current clinical diagnosis of mild cognitive impairment (MCI) or dementia (minimum Blind-Montreal Cognitive Assessment (Blind-MoCA) score of 13/22 conducted over the telephone. The exclusion criteria included drug or alcohol dependence, visual impairment, stroke, brain trauma, a major physical disability that may impact physical activity, intellectual disability that may impact task performance, or cancer treatment requiring chemotherapy in the last five years [45].

### 2.3. Assessment of Diet

The Australian Eating Survey (AES) Food Frequency Questionnaire was self-completed on an iPad under the supervision of a research assistant. The AES asks about the frequency of consumption of 120 food items over the last six months. Participants rate consumption frequency from ‘*Never*’ to ‘*4 or more times per day*’ for food items and beverages up to ‘*7 or more glasses per day*’. Nutrient intake was quantified based on the AUSNUT 2011–2013 database [46]. AES has strong reliability and validity when evaluated against 3-day weighed food records [47]. In addition, the AES has been validated for fruit and vegetable intake using plasma carotenoid concentrations (biomarker for fruit and vegetable intake) [48] and skin carotenoids [49].

As the AES does not collect data on dietary oil intake, a supplementary survey was used to record the consumption frequency of different types of oils, butter, lard, and margarine, based on Swierk et al. [50]. Participants were asked to quantify their daily consumption of culinary fats and oils using a Likert response scale, including 10 options ranging from ‘*Less than 1/2 tablespoons*’ to ‘*8 or more tablespoons’*. Likert scale responses from both surveys were converted to the average weekly serves and the standardised serves were used in the PCA.

### 2.4. Assessment of Cognition: Cambridge Neuropsychological Test Automated Battery

Five tests from the Cambridge Neuropsychological Test Automated Battery (CANTAB) were used to evaluate cognitive function across four key cognitive domains relevant to ageing. The CANTAB is a computerised test battery that has been used to differentiate between clinical populations and healthy controls and correlates moderately with traditional neuropsychological tests in both younger and older populations [51,52]. CANTAB tests have test-retest reliability ranging from weak to strong (0.56 to 89, see Appendix A for more information) [53,54]. The following CANTAB tests were conducted assessing short-term memory (paired associates learning, PAL; immediate recall and recognition; verbal recognition memory, VRM), and long-term memory (delayed recognition; VRM), processing speed (reaction time test, RTI), and executive function (multitasking test, MTT; One Touch Stockings of Cambridge, OTS). To ensure consistency in cognitive testing conditions, assessments were conducted by trained research assistants in a quiet, temperature-controlled office. The sequence of assessments was largely consistent across participants to reduce potential confounds from order and time of day effects. The administration of the full suite of CANTAB tests, inclusive of a 5-min familiarisation task, requires approximately 40 min per participant.

### 2.5. Cognitive Domain Variable Derivation

For tests where lower scores represent better performance (e.g., reaction time), the scores were reversed. The raw scores from the CANTAB tests were standardised to z-scores using the sample mean and standard deviation. These z-scores were then aggregated into cognitive composites based on the Cattell–Horn–Carroll–Miyake (CHCM) cognitive domain taxonomy. The CHCM taxonomy guided the grouping of individual test z-scores into broader cognitive domains: Processing speed, executive function, short-term memory, and long-term memory, as described by Mellow et al. [55] (see Table 1).

### 2.6. Metabolic Syndrome Severity Score (MetSSS)

The MetSSS is a standardised tool designed to quantify the severity of cardiometabolic risk, providing an overall indication of the risk of developing metabolic syndrome [44]. It is a composite score derived from measures of waist circumference (centimetres), triglycerides (millimoles per litre), high-density lipoprotein (HDL) cholesterol (millimoles per litre), systolic and diastolic blood pressure (millimetres of mercury Hg), and blood glucose (millimoles per litre). The MetSSS score is continuous with a value of zero indicating that the participant’s values for all metabolic syndrome risk factors are below the clinical thresholds of concern, suggesting a lower likelihood of developing metabolic syndrome. The ‘pscore’ R package was used to formulate a continuous MetSSS variable [56]. For visualisation purposes, the MetSSS was broken down into four groups, based on the spread of scores.

### 2.7. Variables That Contributed to the MetSSS

Anthropometry: Anthropometric measures were taken twice and completed in accordance with the International Standards for Anthropometric Assessment [57]. Height, in metres, was determined via a wall-mounted stadiometer, and weight scales (TANITA BC-418 Bioelectrical Impedance Analysis Scales, Kewdale, Western Australia, Australia) were used to assess weight in kilograms. Weight and height data were then used to calculate the Body Mass Index (BMI). A 2 m metal measuring tape was used to assess waist circumference.

Blood pressure: Three systolic and diastolic blood pressure measures were collected using an appropriately sized Omron Blood Pressure monitor cuff placed on the left arm. The mean and standard deviation were calculated. Blood pressure measures were taken after participants were seated for at least 5 min and had not consumed caffeine or tobacco in the previous 60 min leading up to the measurement. Readings were taken at least 1 min apart.

Blood collection and analysis: Fasted blood samples were obtained via venepuncture, centrifuged at 4000 rpm for 10 min, and the plasma was aliquoted and stored at −80 °C until further analysis. Analytical assessments of total cholesterol, triglycerides, HDL, high sensitivity C-Reactive Protein (hsCRP), and glucose were performed using the KONELAB 20XTi auto-analyser (Thermo Fisher, Waltham, MA, USA) in conjunction with specific Thermo Fisher reagents. Low-density lipoprotein (LDL) cholesterol levels were calculated using the equation LDL = (total cholesterol − HDL) − (trig/2.17). Prior to the analysis, samples were thawed on ice, vortexed, and then centrifuged at 10,000 rpm for 2 min to eliminate particulate matter. The instrument was calibrated using appropriate calibrators tailored for each test (provided by Thermo Fisher). For each test run, 150 µL of the sample was carefully transferred into a sample cup, ensuring the absence of bubbles. These samples were then introduced to the autoanalyzer, complemented by the necessary Thermo Fisher reagents, and were analysed based on light absorbency measurements.

### 2.8. Covariates

Age and sex data were obtained from a demographic questionnaire, while total years of education were derived from the Australian National University Alzheimer’s Disease Risk Index (ANU-ADRI) [58]. Total daily kilojoules (energy intake) were estimated using the AES. Time spent in moderate-vigorous physical activity (MVPA) was assessed using accelerometry. Participants wore a triaxial accelerometer (Axivity AX3) on their non-dominant wrist continuously for seven days, capturing data at 100 Hz. Raw data were processed with the Open Movement GUI (OmGUI) software V1.0.0.45 and a custom MATLAB R2018B interface (COBRA). For further details, refer to Mellow et al. [55].

### 2.9. Principal Component Analysis (PCA) to Derive Dietary Patterns

The 134 food items from the Australian Eating Survey (AES) and oils questionnaire were reduced into 40 food groups (see Appendix A for more details) based on nutrient profile, culinary use, and prior research that also involved older Australians [25,27]. When food items did not clearly fit into a group, they were left as standalone items (e.g., avocado has a macronutrient profile that is dissimilar to other vegetables). The categorisation process was guided by nutrition experts (AW) and dietitians (CC, KM). Following this, items were removed if over 85% of the population reported not consuming the food item in the past 6 months (a similar method was described in Jacka et al. [28]).

Prior to the PCA, the Kaiser Meyer Olkin (KMO; [59]) test was conducted to assess the adequacy of the sample size. Following the PCA, a post-estimation KMO test was applied to each individual food group, all of which met the cut-off value of 0.5, indicating that the sample size was sufficient for PCA analysis. Extreme values of the additional oils questionnaire (olive oil group, butter and lard group, margarine group, and other oils group) were truncated by using empirical quantiles at 0.05 and 0.95 to lessen the influence of extreme values and meet the assumptions of PCA.

PCA was used to reduce the food group variables into a smaller number of principal components representing dietary patterns. PCA steps were conducted using the ‘Psych’ package in R [60]. The components involve a linear combination of the original variables that represent the majority of the sampled dietary variance. A varimax (orthogonal) rotation was implemented to maximise the variance of loadings within each component. This method increases the distinctiveness of dietary patterns and improves the interpretability of loadings, crucial for analyses in nutritional epidemiology (see examples [25,27,28,61]). A sensitivity analysis using oblimin rotation was also completed to ensure that the identified patterns were robust (see Appendix A).

The selection of principal components was guided by evaluating the percentage of total variance, as indicated by a scree plot of eigenvalues, the eigenvalue magnitudes, and the interpretability of loadings (i.e., what appeared to be meaningful dietary patterns and not redundant). Using a multifaceted criterion provides a balanced approach to ensure that the retained components adequately summarise the original data while avoiding overfitting [62]. Loadings under 0.1 are presented but were not included in the calculation of principal component scores. This threshold was to ensure that only meaningful loadings contribute to the dietary pattern interpretation.

### 2.10. Removal of Participants with Implausible Energy Intake

Self-report diet measurement tools may lead to misreporting of energy consumption [63]. To minimise the risk of false inference, individuals who over-estimated or under-estimated their kilojoule intake beyond reasonable thresholds were removed from the final analysis. Unreasonable self-reported energy intake values were defined using cut-offs (<2092 kJ or >14,644 kJ) for women and (<3347 kJ or >17,573 kJ) for men [24]. In total, seven participants (4 females; 3 males) were removed.

### 2.11. Statistical Analysis

All statistical analyses were performed using R version 4.3.0 [64]. Overall means were presented, and categorical variables were displayed in percentages. Preliminary t-tests were used to compare mean values between males and females to determine the inclusion of sex in the model adjustment. The significance level was set at *p* < 0.05, 2-tailed (Table 2). Pearson’s correlation (or Spearman’s Rho for non-normally distributed variables) was used to examine how dietary patterns correlated with key demographic, cardiometabolic, and cognitive variables. The data were first visually inspected for normality.

Missing data were addressed using multiple imputations with chained equations (MICE) as implemented in the ‘MICE’ R package [65]. Imputation was conducted using predictive mean matching as the methodological approach to estimate missing values. Five imputed data sets were created, and a random seed was set to ensure the reproducibility of the results. Additionally, a sensitivity analysis was planned to be conducted using the raw dataset, excluding participants with imputed data, to verify the robustness of the results.

Associations between dietary pattern scores and performance in each of four cognitive domains (z-score; outcome variable) were examined using a series of multiple linear regression models. For each cognitive domain, three models were considered: Model 1 had the three dietary patterns derived from PCA as predictors; Model 2 was adjusted to control for age, sex, and education level (included as there is evidence to suggest that education can attenuate the association between dietary patterns and cognition [24]; and Model 3 also controlled for energy intake and moderative-to-vigorous physical activity (MVPA).

The same three-model approach was used to test whether cardiometabolic risk status (MetSSS) moderated the relationship between dietary alignment scores and the four cognitive domain scores. Here, Model 1 included only the dietary patterns, MetSSS, and three separate interaction terms (one for each diet with MetSSS). Model 2 introduced age, sex, and education. Model 3 also added daily energy intake and MVPA. Residuals were plotted against fitted values to assess linearity. The significance level was set at α = 0.05. Given the exploratory nature of this study, particularly in the use of PCA to identify dietary patterns, *p*-values were not adjusted for multiple comparisons.

## 3. Results

### 3.1. Participant Characteristics

Data from 417 participants, including 125 males and 292 females, were included in the analysis (Figure 1). The mean age was 65.53 years (SD = 2.96). On average, participants had 15.35 (SD = 3.31) years of formal education, 72% of participants were married, and 78% of participants were born in Australia.

The average BMI was in the overweight category (26.95 kg/m^2^), with 24.5% of the sample categorised as having obesity (BMI > 30 kg/m^2^), as defined by the World Health Organisation. Males exhibited significantly higher LDL cholesterol and waist circumference, nearing clinical thresholds for high risk, while females had significantly higher total and HDL and total cholesterol, alongside lower systolic blood pressure (Table 2).

### 3.2. Dietary Patterns

After initial data processing that grouped food items based on nutrient profiles and culinary uses, ‘other milks’ and ‘liver’ were removed as 85% of respondents reported never consuming these food items, leaving 38 groups for the PCA.

Based on the dominant loadings of each component, the principal components were given representative dietary pattern labels (see Table 3). Principal component 1 (explaining 11.4% of the variance) was characterised by high positive loadings for vegetables, legumes, and dairy, along with negative loadings for meat consumption, and was thus labelled a plant-dominant diet. Principal component 2 (explaining 8.2% of the variance) was characterised by high loadings of meat but was also low on plant-based foods, so it was labelled a meat-dominant diet. Principal component 3 (explaining 7.2% of the variance) was characterised by high loadings of discretionary foods, processed foods, foods high in simple sugars, and low loadings in healthful foods, leading to the label of a Western-style diet. Together, these three principal components explained 26.8% of the total variance in the food group data. 

Dietary pattern scores ranged from −12.02 to 15.64 for the plant-dominant diet, −7.60 to 14.67 for the meat-dominant diet, and −7.80 to 11.93 for the Western-style diet. On average, females tended to have significantly higher scores on the plant-dominant diet than males, and males tended to score significantly higher on both the meat-dominant diet and the Western-style diet compared to females (Table 4).

Correlations revealed that a higher plant-dominant diet score was correlated with lower MetSSS, BMI, waist circumference, triglycerides, and hsCRP (Table 5). A higher plant-dominant diet score was also correlated with higher levels of HDL cholesterol and total cholesterol, which may have been driven by the higher HDL. A higher plant-dominant diet was significantly correlated with higher MVPA. A higher meat-dominant diet score was correlated with a higher MetSSS, BMI, waist circumference, triglycerides, systolic and diastolic blood pressure, and higher hsCRP. It was also correlated with lower HDL and more time spent in MVPA.

The Western-style diet showed a similar pattern to the meat-dominant diet, with greater Western-style diet alignment being significantly correlated with higher levels of MetSSS, BMI, triglycerides, and waist circumference, but significantly lower HDL. In addition, it correlated with higher hsCRP and systolic blood pressure. The meat-dominant diet was positively correlated with the Western-style diet. In contrast, both these diets showed weaker and negative correlations with the plant-dominant diet. Variance inflation factor (VIF) analysis indicated no major concerns regarding multicollinearity among the variables, with a slight increase in energy intake, as detailed in Appendix A.

### 3.3. Dietary Patterns and Cognitive Composite Outcomes

Table 6 summarises results from multiple linear regression models, examining associations between the three dietary patterns and composite scores for each cognitive domain. As detailed in the Methods, for each cognitive domain, Model 1 only included dietary patterns; Model 2 adjusted for age, sex, and education; and Model 3 further adjusted for energy intake and MVPA. The plant-dominant diet did not show any statistically significant associations with any cognitive outcomes.

Greater alignment to the meat-dominant diet was associated with poorer long-term memory (Model 1: β = −0.04, 95% CI [−0.08, −0.004], *p* = 0.028). This negative association persisted when controlling for age, sex, and education (Model 2; β = −0.04, 95% CI [−0.08, −0.004]), *p* = 0.030) as well as energy intake and MVPA (Model 3; β = −0.04, 95% CI [−0.08, −0.004], *p* = 0.046). The Western-style diet was associated with a higher long-term memory (Model 1; β = 0.04, 95% CI [0.006–0.08], *p* = 0.042) and when controlling for age, sex, and education (Model 2; β = 0.05, 95% CI [0.01, 0.09], *p* = 0.027).

None of the dietary patterns were significantly associated with short-term memory, executive function or processing speed.

### 3.4. Sensitivity Analysis Without Imputed Data

When using the data without imputation, there was some evidence that the meat-dominant diet was associated with poorer executive function (see Appendix A). Model 1 and Model 2 showed no significant associations between executive function scores with any dietary patterns, whereas Model 3 showed a marginal negative association between the meat-dominant diet and executive function (β = −0.02, 95% CI [−0.03, 0.03], *p* = 0.050).

Greater alignment with the meat-dominant diet was associated with slower processing speed (Model 1; β = −0.03, 95% CI [−0.06, −0.002], *p* = 0.038). This association persisted when controlling for age, sex, and education (Model 2; β = −0.03, 95% CI [−0.06, −0.001], *p* = 0.042) but not when also controlling for MVPA (β = 0.001, 95% CI [0.001, 0.004], *p* = 0.001) and energy intake (β = 0.002, 95% CI [−0.05, 0.02], *p* = 0.859). However, unexpectedly, the Western-style diet was associated with higher processing speed in Model 3 (β = 0.05, 95% CI [0.006, 0.09], *p* = 0.024), suggesting that more consumption of discretionary foods is associated with faster processing speed.

### 3.5. The Moderating Effect of Cardiometabolic Health

Additional analyses were conducted to examine whether metabolic health (i.e., MetSSS) moderated the relationship between dietary patterns and cognitive outcomes (Table 7), using the same structure of models as above.

**Long-term memory:** The relationship between the meat-dominant diet and long-term memory was significantly moderated by MetSSS (Model 1; β = −0.03, *p* = 0.011, 95% CI [−0.05, −0.01]), and this moderating effect remained significant when controlling for age, sex, and education (Model 2; β = −0.03, *p* = 0.012, [−0.05, −0.01]) and energy intake and MVPA (Model 3; β = −0.03, *p* = 0.015, [−0.05, −0.01]).

To visualise this moderation effect, participants were divided into four groups based on their MetSSS score: 0 (indicating no metabolic risk factors, n = 67), 1–2 (low to moderate metabolic risk, n = 165), 3–5 (moderate to high metabolic risk, n = 158), and 5+ (high metabolic risk, n = 34). As shown in Figure 2, the relationship between the meat-dominant diet and long-term memory was most evident in people with a higher metabolic risk (groups 3 and 4).

**Executive function:** Although earlier analyses showed no association between executive function and any dietary pattern, MetSSS significantly moderated the plant-dominant diet only when controlling for age, sex, and education (Model 2: β = 0.01, *p* = 0.029, [0.001, 0.01]), as well as energy intake and MVPA (Model 3: β = 0.01, *p* = 0.037, [0.0005, 0.01]).

Figure 3 shows that the relationship between executive function and a plant-dominant diet varied by MetSSS level. Specifically, for people with high metabolic risk, greater alignment to a plant-dominant diet was associated with higher executive function scores. In contrast, and quite unexpectedly, for people with low metabolic risk, greater alignment to a plant-dominant diet is associated with lower executive function scores.

There were no significant moderation effects of MetSSS on the relationship between dietary patterns and either processing speed or short-term memory.

### 3.6. Sensitivity Analysis Without Imputed Data for Moderation Analysis

To confirm that the multiple imputation did not bias the findings, these analyses were repeated with the raw dataset. All interactions remained stable in direction, magnitude, and significance and as such, have not been reported here (see Appendix A).

## 4. Discussion

The current study examined associations between dietary patterns and cognitive outcomes in a sample of healthy older adults. The principal component analysis identified three main dietary patterns: plant-dominant, meat-dominant, and Western-style, and examined their relationship with four cognitive domains: long-term memory, short-term memory, executive function, and processing speed. It was determined that there was one healthful dietary pattern and two unhealthful patterns. Results demonstrated that while there were no positive associations with the healthful dietary pattern (i.e., the plant-dominant diet), the meat-dominant diet was associated with poorer long-term memory performance. There was also evidence to suggest that differing profiles of cardiometabolic health influenced this relationship.

### 4.1. Meat-Dominant Diet and Cognition

The meat-dominant diet had a small, negative association with long-term memory performance. This is aligned with research that has reported that lower meat intake is associated with faster processing speed, higher verbal ability, better overall cognition, and larger brain volume [13,15,66]. These associations remained significant after adjusting for demographic variables such as age, sex, total years of education, energy intake, and MVPA.

The association between the meat-dominant diet and long-term memory was more pronounced for people with high cardiometabolic risk. In contrast, there was little association between the meat-dominant dietary pattern and long-term memory amongst people with low metabolic risk. High meat intake tends to be associated with higher saturated fatty acid intake [67]. While consumption of polyunsaturated fatty acids has been documented to be protective against dementia, greater saturated fat intake is associated with cognitive decline and dementia [68,69,70,71,72,73].

Greater consumption of saturated fats is also linked to elevated plasma cholesterol and thereby risk for the development of atherosclerosis (see review [74]). Poor cardiometabolic health outcomes are associated with a higher risk of cognitive decline and dementia [37,38,39]. Therefore, individuals with the highest metabolic risk may experience the poorest cognitive outcomes when consuming a meat-dominant, suboptimal diet. The moderation effect of the meat-dominant diet on long-term memory may also suggest a genetic predisposition that may influence individual susceptibility to dietary effects (see review [75]). Additionally, the absence of essential nutrients rich in plant-based foods, including antioxidants, flavonoids, carotenoids, and dietary fibre, may also explain these associations. When using raw data (i.e., without imputation) in a sensitivity analysis, a similar relationship was observed where a higher meat-dominant diet score was also associated with lower processing speed.

### 4.2. Western-Style Diet and Cognition

Our data did not replicate previously reported negative associations between a Western-style diet and cognitive function (see review [26]). Unexpectedly, alignment with a Western-style diet was associated with a significantly higher score in long-term memory. However, this effect did not persist when controlling for variables related to energy expenditure (MVPA and energy intake). It is possible that the short-term benefits of high glycaemic load diets on alertness may have influenced these results, supported by studies that report better cognitive performance following acute oral glucose administration [76,77]. This is because glucose can facilitate cognitive performance by providing an immediate source of energy for brain function [78].

### 4.3. Plant-Dominant Diet and Cognition

Primary analyses found no statistically significant positive association between a plant-dominant diet and any cognitive domain. Previous research has demonstrated that diets, such as the MedDiet, rich in vegetables and healthy fats, coupled with moderate alcohol intake, are associated with better cognitive performance and reduced dementia risk [10,11]. The absence of such findings may indicate that the cognitive effects of the plant-dominant dietary pattern are subtle or influenced by other factors, such as protein or other nutrient intakes or a generally healthy lifestyle. Further research with a longitudinal design could provide more insights into the potential cognitive benefits of this dietary pattern. Lower levels of foods containing vitamin B12, only sourced via animal protein foods, have been associated with poorer cognitive health (see review [79]). This could also be one potential explanation for these findings.

The interaction between the plant-dominant diet and MetSSS presents a novel finding concerning executive function. While the plant-dominant diet was not initially associated with any cognitive variables, MetSSS significantly moderated the relationship between the plant-dominant diet and executive function. In individuals with low metabolic risk, slight improvements in executive function are observed with increased adherence to a plant-dominant diet, whereas those at high metabolic risk exhibit lower levels of executive function when their plant-dominant diet adherence scores are higher.

### 4.4. Strengths, Limitations, and Future Directions

Previous studies have not found a moderating effect of vascular risk factors or comorbidities on the relationship between dietary patterns (i.e., MedDiet) and cognition [25,80,81]. These papers have introduced a final adjustment model to control for the presence or absence of cardiometabolic conditions. This simplistic approach may not capture the multifaceted nature of metabolic health and its impact on cognitive function. The current paper used a metabolic risk score to quantify the effect of metabolic health status on the relationship between dietary patterns and cognitive outcomes. This methodology provides a more nuanced understanding of the interplay between diet, metabolic health, and cognitive function.

Another strength of the current paper was the use of PCA to quantify whole dietary patterns, providing a rich and multidimensional measure of dietary pattern alignment in our sample. In nutritional research, dietary patterns are commonly determined using two primary methodologies: a priori and a posteriori [82]. A priori methods rely on predefined criteria to quantify an individual’s alignment to a specific dietary pattern. However, these methods can be overly simplistic and may not capture the complexity of various food group pairings. Alternatively, dietary pattern alignment may be quantified through data-driven a posteriori approaches, such as this paper has done. PCA examines sample-specific consumption patterns by condensing multiple food items into a small number of components that account for the greatest variation in the data [4]. PCA can uncover multidimensional dietary patterns, moving beyond a simple diet quality spectrum that accurately reflects the real-world complexity of dietary behaviours, offering a more granular understanding of how specific dietary configurations correlate with health outcomes.

A primary limitation of this research is its cross-sectional design, which precludes the ability to establish causality and discern temporal relationships between dietary patterns and outcomes. The absence of longitudinal data limits our capacity to understand how these relationships might develop or change over time. This is particularly important as the effects of poor metabolic health are likely to be influenced by the magnitude and duration of cumulative exposure. Moreover, as the cohort was largely healthy and active, this may have prevented the detection of greater effect sizes.

While this study provides key insights into the associations between diet, metabolic health, and cognitive function within an older Australian cohort primarily of Caucasian ethnicity, it is important to note that dietary patterns and their implications can vary significantly across different cultural groups and time periods. To enhance the global applicability of our results, further studies are needed across diverse geographical and cultural settings. Additionally, the use of PCA in this study was exploratory, and as such, p-values were not adjusted for multiple comparisons, which may increase the likelihood of Type I errors.

Lastly, the use of food frequency questionnaires to assess dietary patterns carries inherent limitations due to reliance on participant recall. In the future, utilising objective nutrient biomarkers in conjunction with self-report measures may help overcome the limitations associated with self-reported nutritional data. However, a strength was the use of the Australian Eating Survey which has been validated against other dietary assessments and nutrient intakes [47,48]. Moreover, the inclusion of blood-based metabolic biomarkers (such as plasma total cholesterol, HDL, LDL, and hsCRP) and their correlation with dietary pattern scores was another strength.

## 5. Conclusions

In conclusion, the current findings contribute to the growing evidence of the importance of overall dietary patterns for cognitive health in older adults. The associations observed in this study between dietary patterns and cognitive outcomes, though small, are noteworthy. A particularly novel finding was that the cardiometabolic health profile moderated the relationship between the meat-dominant diet with long-term memory performance and the plant-dominant diet with executive function. Together, these findings are aligned with the growing scientific consensus that nutrition should be tailored and personalised, with important implications for public health and nutritional guidance [83]. Given the cross-sectional nature of our research, these modest associations could potentially underrepresent the true impact of long-term dietary habits and cumulative metabolic risk on cognitive health. This study lays the groundwork for future longitudinal research that is critical to fully understanding the nuanced effects of diet and cardiometabolic health on cognitive ageing across varied demographic landscapes.

## Figures and Tables

**Figure 1 nutrients-16-03890-f001:**
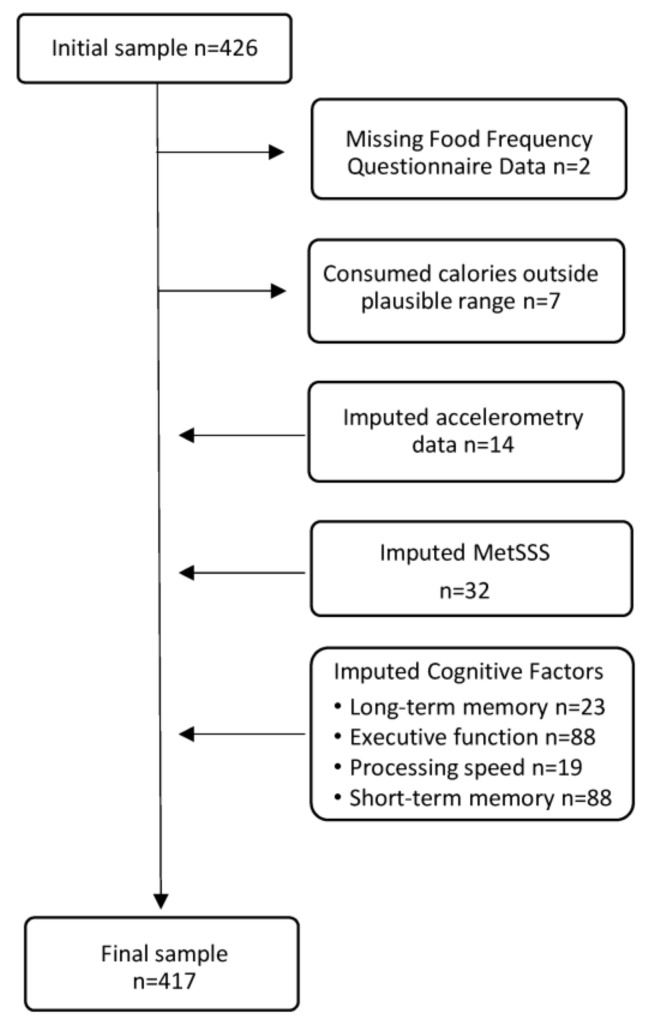
Flowchart illustrating participants excluded due to missing data. The initial sample consisted of 426 participants. Participants with implausible daily energy intake, defined as less than 2092 kJ for females and 3347 kJ for males or more than 14,644 kJ for females and 17,573 kJ for males, were excluded. Remaining missing data were imputed.

**Figure 2 nutrients-16-03890-f002:**
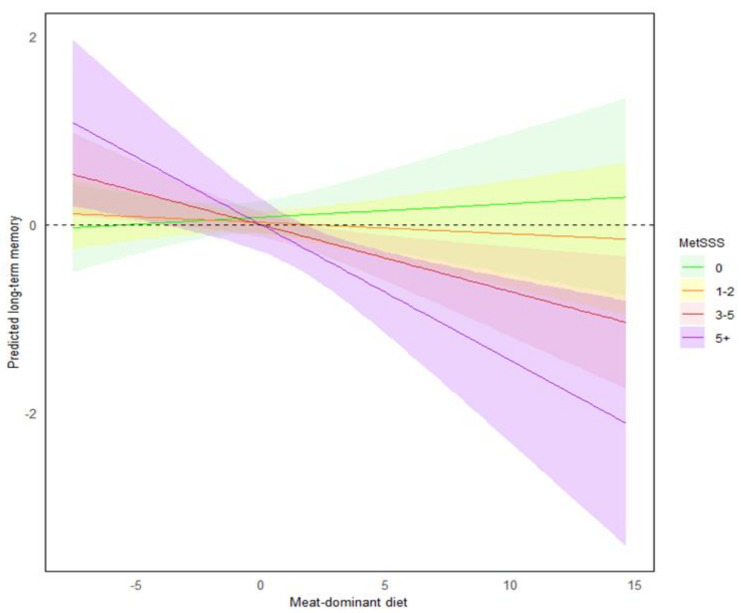
The relationship between long-term memory and the meat-dominant diet pattern was moderated by the level of metabolic risk. To present the moderating effect of MetSSS, participants were allocated into four groups based on their MetSSS score with 0 representing no risk factors and 5+ indicating high risk. The meat-dominant diet was associated with poorer long-term memory in people with higher metabolic risk only. A more positive score on the *Y*-axis indicates better cognitive performance and, on the *X*-axis, indicates greater alignment with the diet score. The shaded regions represent the 95% confidence intervals around the predicted values. This visualisation is based on the fully adjusted regression model (Model 3) that controls for age, sex, total years of education, energy intake, and MVPA.

**Figure 3 nutrients-16-03890-f003:**
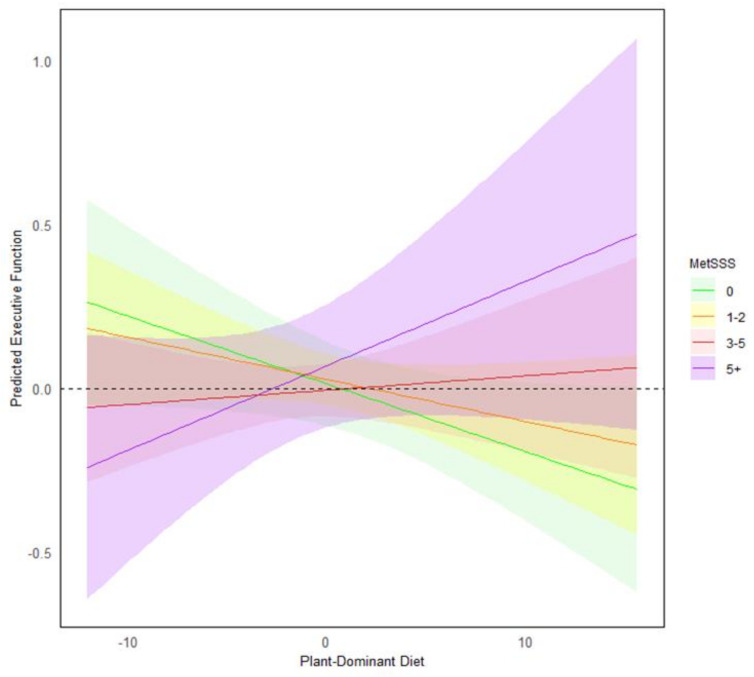
Predicted executive function based on different levels of plant-dominant diet intake across varying categories of MetSSS. To aid in interpretation, the moderating effect of MetSSS is depicted by allocating participants into four groups based on their MetSSS score (0 represents no risk factors and 5+ indicates high risk). Participants with the highest metabolic risk have higher executive function scores when they have greater alignment with the plant-dominant diet.

**Table 1 nutrients-16-03890-t001:** Scores from the CANTAB tests were used to construct each cognitive domain composite. This table has been adapted from Mellow et al. [55]. Asterisks indicate scores that were reversed so that higher scores denote better performance for all measures.

Cognitive Domain	Task	Outcome Measures
Long-term memory	Verbal Recognition Memory	Delayed recognition—total correct
Short-term memory	Verbal Recognition Memory	Immediate recognition—total correct Immediate free recall—total correct distinct words
	Paired Associates Learning	Total errors (adjusted) * First attempt memory score
Executive function	Multitasking test	Total incorrect responses * Median response latency multitasking cost * Median response latency incongruency cost *
	One Touch Stockings of Cambridge	Problems solved on first choice
Processing speed	Reaction time	Median latency to first choice * Median simple reaction time * Median 5-choice reaction time * Median simple movement time * Median 5-choice movement time *

**Table 2 nutrients-16-03890-t002:** Cardiometabolic health indicators and clinical cut-offs for males and females. *p*-values indicate sex differences for each variable using t-tests.

Indicator	Overall (SD)	Female (SD)	Male (SD)	*p*-value	Clinical Cut-Off
Age	65.53 (2.96)	65.37 (2.85)	65.91 (3.19)	0.106	
BMI (kg/m^2^)	26.95 (5.07)	26.7 (5.36)	27.53 (4.26)	0.099	BMI > 25 = overweight BMI > 30 = obese
LDL cholesterol (mmol/L)	3.2 (0.99)	3.27(0.99)	3.02 (0.98)	0.020	≥3.37 mmol/L = high
HDL cholesterol (mmol/L)	1.76 (0.51)	1.88 (0.49)	1.48 (0.44)	<0.001	Males: <1.03 mmol/L = low Females: <1.29 mmol/L = low
Total cholesterol (mmol/L)	5.50 (1.06)	5.68 (1.02)	5.10 (1.03)	<0.001	≥5.5 mmol/L = high
Glucose (mmol/L)	5.10 (0.70)	5.05 (0.69)	5.22 (0.70)	0.051	Prediabetes: 5.6 to 6.9 mmol/L Diabetes: ≥7.0 mmol/L
Triglycerides (mmol/L)	1.18 (0.77)	1.13 (0.52)	1.29 (1.17)	0.142	>1.70 mmol/L = high
Waist circumference (cm)	89.70 (14.40)	85.82 (13.63)	98.86 (12.10)	<0.001	High risk: ≥88 cm for women, ≥102 cm for men
High sensitivity C-reactive protein (mg/L)	1.76 (3.21)	1.92 (3.54)	1.38 (2.23)	0.061	>3 mg/L considered high
Systolic blood pressure (mmHg)	134 (18)	131 (17.32)	143 (16.48)	<0.001	High ≥ 130 mmHg Hypertense > 140 mmHg
Diastolic blood pressure (mmHg)	81 (10)	80 (10.17)	82 (11)	0.080	High ≥ 90 mmHg Hypertense > 100 mmHg

**Table 3 nutrients-16-03890-t003:** Food group loadings for principal components 1, 2, and 3, where blue indicates a positive loading (high consumption) and red indicates a negative loading (low consumption).

Food Groups	Plant-Dominant Diet	Meat-Dominant Diet	Western-Style Diet
Other vegetables	0.75	0.1	−0.11
Red–yellow vegetables	0.72	0.03	0.15
Cruciferous vegetables	0.66	0.01	−0.1
Green leafy vegetables	0.66	0.12	−0.16
Legumes	0.62	−0.19	0.13
Nuts	0.57	−0.2	0.12
Avocado	0.54	0.03	−0.28
Fruit	0.51	−0.09	0.23
Seafood	0.34	0.19	−0.31
Dried fruit	0.33	−0.19	0.36
Dairy normal	0.33	0.22	0.19
Water	0.31	−0.06	−0.14
Eggs	0.29	0.42	−0.19
Olive oil	0.25	0.24	−0.21
Soup	0.24	−0.03	0.17
Condiments	0.24	0.4	0.16
Whole grains	0.2	0.12	0.11
Refined grains	0.18	0.03	0.51
Other oils	0.16	0.21	0.07
Chicken (served with veg)	0.14	0.43	−0.23
Tea/coffee	0.09	0.03	−0.13
Potato	0.07	0.16	0.46
Butter	0.06	0.38	−0.05
Sweet snacks	0.04	0.25	0.66
Chocolate	0.04	0.02	0.41
Canned fruit	0.03	−0.06	0.25
Meat (served with vegetables)	0	0.53	−0.09
Margarine	−0.03	0.25	0.32
Discretionary dairy	−0.04	0.35	0.15
Processed meat	−0.05	0.54	0.25
Alcohol	−0.08	0.23	−0.15
Savoury snacks	−0.11	0.38	0.35
Sugary beverages	−0.13	0.15	0.41
Chicken (without veg)	−0.13	0.47	0.11
Meat (without veg)	−0.14	0.66	0.21
Diet soft drink	−0.21	0.15	0.08
Fried protein	−0.27	0.4	0.31
Fast-fried food	−0.29	0.44	0.45

Note. Loadings under 0.1 are presented but were not included in the final creation of principal component scores.

**Table 4 nutrients-16-03890-t004:** Means and standard deviations for dietary patterns overall and by sex using t-tests.

Diet	Mean (SD)	Females (SD)	Males (SD)	*p*-value
Plant-dominant diet	−0.10 (4.27)	0.59 (4.26)	−1.70 (3.84)	<0.001
Meat-dominant diet	−0.08 (3.05)	−0.27 (2.9)	0.37 (3.33)	0.040
Western-style diet	−0.05 (2.79)	−0.35 (2.68)	0.67 (2.93)	0.001

**Table 5 nutrients-16-03890-t005:** Correlation coefficients between dietary patterns and demographics, cardiometabolic health, and cognitive health scores.

Variable		Plant-Dominant Diet	Meat-Dominant Diet	Western-Style Diet
**Dietary patterns**				
Plant-dominant diet	r		−0.16	−0.21
	*p*		*0.001*	*<0.001*
Meat-dominant diet	r			0.49
	*p*			*<0.001*
**Demographics**				
Education	r	0.06	−0.04	−0.07
	*p*	*0.214*	*0.379*	*0.139*
Total MVPA	r	0.15	−0.14	−0.07
	*p*	*0.003*	*0.006*	*0.142*
Energy intake	r	0.27	0.50	0.57
	*p*	*<0.001*	*<0.001*	*<0.001*
**Cardiometabolic health**				
MetSSS	r	−0.27	0.29	0.21
	*p*	*<0.001*	*<0.001*	*<0.001*
BMI	r	−0.22	0.34	0.17
	*p*	*<0.001*	*<0.001*	*<0.001*
LDL	r	0.04	0.00	−0.09
	*p*	*0.396*	*0.985*	*0.090*
HDL	r	0.26	−0.11	−0.21
	*p*	*<0.001*	*0.024*	*<0.001*
Total cholesterol	r	0.10	−0.02	−0.16
	*p*	*0.028*	*0.658*	*0.002*
Triglycerides	r	−0.12	0.16	0.16
	*p*	*0.016*	*0.001*	*0.002*
Waist	r	−0.33	0.32	0.23
	*p*	*<0.001*	*<0.001*	*<0.001*
Glucose	r	−0.08	0.17	0.07
	*p*	*0.112*	*0.001*	*0.166*
hsCRP	r	−0.10	0.15	0.19
	*p*	*0.047*	*0.003*	*<0.001*
Systolic blood pressure	r	−0.15	0.15	0.13
	*p*	*0.003*	*0.002*	*0.009*
Diastolic blood pressure	r	−0.02	0.13	0.09
	*p*	*0.632*	*0.007*	*0.061*
**Cognitive Composites**				
Long-term memory	r	0.06	−0.08	0.04
	*p*	*0.208*	*0.114*	*0.482*
Short-term memory	r	0.05	0.03	0.06
	*p*	*0.400*	*0.562*	*0.240*
Processing speed	r	0.06	−0.07	0.03
	*p*	*0.242*	*0.160*	*0.560*
Executive function	r	−0.05	−0.05	0.04
	*p*	*0.289*	*0.306*	*0.467*

Note. Triglycerides and high-sensitivity C-reactive protein (hsCRP) correlations were calculated using Spearman’s Rho. BMI = body mass index; LDL = low-density lipoprotein; HDL = high-density lipoprotein; MetSSS = metabolic syndrome severity score; MVPA = moderate-to-vigorous physical activity.

**Table 6 nutrients-16-03890-t006:** Regression outcomes for cognitive functions across different dietary patterns and three progressive model adjustments. Model 1 includes only dietary patterns; Model 2 adjusts for age, sex, and education; and Model 3 also adjusts for energy intake and physical activity.

			Plant-Dominant Diet	Meat-Dominant Diet	Western-Style Diet	Age	Sex	Education	Energy Intake	MVPA
**Long-term** **Memory**	Model 1	β	0.02	**−0.04**	**0.04**					
		T	1.28	**−2.22**	**2.04**					
		p	0.202	**0.028**	**0.042**					
		95% CI	[−0.01, 0.04]	**[−0.08, −0.00]**	**[0.00, 0.08]**					
	Model 2	β	0.01	**−0.04**	**0.05**	−0.02	0.18	0.03		
		T	0.85	**−2.19**	**2.27**	−1.00	1.54	1.98		
		p	0.396	**0.030**	**0.024**	0.320	0.124	0.050		
		95% CI	[−0.01, 0.04]	**[−0.08, −0.00]**	**[0.01, 0.09]**	[−0.05, 0.02]	[−0.05, 0.41]	[−0.00, 0.06]		
	Model 3	β	0.00	**−0.04**	0.04	−0.02	0.24	0.03	0.00	0.00
		T	0.23	**−1.98**	1.58	−0.89	1.66	1.95	0.93	0.40
		p	0.815	**0.049**	0.114	0.374	0.099	0.053	0.355	0.691
		95% CI	[−0.03, 0.04]	**[−0.09, −0.00]**	[−0.01, 0.09]	[−0.05, 0.02]	[−0.05, 0.53]	[−0.00, 0.06]	[−0.00, 0.00]	[−0.00, 0.00]
**Short-term memory**	Model 1	β	0.01	−0.01	0.02					
		T	1.15	−0.44	1.52					
		p	0.260	0.669	0.133					
		95% CI	[−0.01, 0.03]	[−0.04, 0.02]	[−0.01, 0.05]					
	Model 2	β	0.01	−0.01	0.02	−0.02	0.04	0.02		
		T	1.01	−0.41	1.61	−1.87	0.56	1.69		
		p	0.319	0.689	0.112	0.068	0.573	0.098		
		95% CI	[−0.01, 0.03]	[−0.04, 0.03]	[−0.01, 0.05]	[−0.04, 0.00]	[−0.10, 0.18]	[−0.00, 0.04]		
	Model 3	β	0.01	0.00	0.03	−0.02	0.01	0.02	0.00	0.00
		T	1.10	−0.27	1.47	−1.90	0.11	1.69	−0.47	−0.39
		p	0.277	0.792	0.147	0.063	0.909	0.100	0.640	0.700
		95% CI	[−0.01, 0.03]	[−0.04, 0.03]	[−0.01, 0.06]	[−0.05, 0.00]	[−0.17, 0.19]	[−0.00, 0.04]	[−0.00, 0.00]	[−0.00, 0.00]
**Executive function**	Model 1	β	−0.01	−0.01	0.01	−	−	−	−	−
		T	−1.04	−1.43	0.97	−	−	−	−	−
		p	0.300	0.154	0.332	−	−	−	−	−
		95% CI	[−0.02, 0.01]	[−0.03, 0.01]	[−0.01, 0.03]	−	−	−	−	−
	Model 2	β	0.00	−0.01	0.01	**−0.04**	**−0.21**	0.01	−	−
		T	−0.18	−1.50	0.59	**−5.19**	**−3.65**	1.82	−	−
		p	0.856	0.135	0.558	**<0.001**	**<0.001**	0.073	−	−
		95% CI	[−0.01, 0.01]	[−0.03, 0.00]	[−0.01, 0.03]	**[−0.06, −0.03]**	**[−0.32, −0.09]**	[−0.00, 0.03]	−	−
	Model 3	β	−0.01	−0.02	0.00	**−0.04**	**−0.17**	0.01	0.00	0.00
		T	−0.76	−1.88	−0.19	**−5.15**	**−2.40**	1.77	1.09	−0.84
		p	0.448	0.060	0.847	**<0.001**	**0.017**	0.080	0.278	0.402
		95% CI	[−0.02, 0.01]	[−0.04, 0.00]	[−0.03, 0.02]	**[−0.06, −0.03]**	**[−0.31, −0.03]**	[−0.00, 0.03]	[−0.00, 0.00]	[−0.00, 0.00]
**Processing speed**	Model 1	β	0.01	−0.02	0.02					
		T	1.10	−1.67	1.51					
		p	0.273	0.097	0.131					
		95% CI	[−0.01, 0.03]	[−0.05, 0.00]	[−0.01, 0.06]					
	Model 2	β	0.01	−0.02	0.02	**−0.03**	−0.12	0.00		
		T	1.47	−1.67	1.28	**−2.72**	−1.36	−0.11		
		p	0.143	0.095	0.201	**0.007**	0.174	0.913		
		95% CI	[−0.00, 0.03]	[−0.05, 0.00]	[−0.01, 0.05]	**[−0.06, −0.01]**	[−0.28, 0.05]	[−0.02, 0.02]		
	Model 3	β	0.02	−0.01	0.04	**−0.03**	−0.14	0.00	0.00	**0.00**
		T	1.62	−0.50	1.94	**−2.58**	−1.28	−0.04	−1.48	**3.24**
		p	0.107	0.619	0.054	**0.010**	0.201	0.971	0.139	**0.001**
		95% CI	[−0.00, 0.05]	[−0.04, 0.02]	[−0.00, 0.08]	**[−0.06, −0.01]**	[−0.35, 0.07]	[−0.02, 0.02]	[−0.00, 0.00]	**[0.00, 0.00]**

**Table 7 nutrients-16-03890-t007:** Multiple regression with interaction terms (with multiple imputations for missing data related to the metabolic syndrome severity score (MetSSS): Model 1 (diet and interaction terms), Model 2 (Model 1 and age, sex, total years of education), and Model 3 (Model 2 and energy intake and MVPA). Diet*MetSSS = interaction term.

			Plant Diet	Meat Diet	Western Diet	MetSSS	Plant *MetSSS	Meat *MetSSS	Western *MetSSS	Age	Sex	Education	Energy Intake	MVPA
**Long** **-** **term Memory**	Model 1	β	0.00	0.02	0.02	−0.03	0.01	−0.03	0.01					
		T	0.08	0.70	0.59	−1.08	1.01	−2.56	1.23					
		*p*	*0.938*	*0.482*	*0.557*	*0.280*	*0.314*	*0.011*	*0.220*					
		95% CI	[−0.03, 0.04]	[−0.04, 0.08]	[−0.04, 0.08]	[−0.09, 0.03]	[−0.01, 0.02]	[−0.05, −0.01]	[−0.01, 0.04]					
	Model 2	β	0.00	0.02	0.02	−0.02	0.01	−0.03	0.01	−0.02	0.16	0.03		
		T	−0.11	0.66	0.69	−0.81	0.98	−2.52	1.26	−1.15	1.35	1.77		
		*p*	*0.915*	*0.511*	*0.493*	*0.418*	*0.329*	*0.012*	*0.208*	*0.251*	*0.179*	*0.078*		
		95% CI	[−0.04, 0.03]	[−0.04, 0.08]	[−0.04, 0.08]	[−0.09, 0.04]	[−0.01, 0.02]	[−0.05, −0.01]	[−0.01, 0.04]	[−0.05, 0.01]	[−0.07, 0.38]	[−0.00, 0.06]		
	Model 3	β	0.00	0.02	0.02	−0.02	0.01	−0.03	0.01	−0.02	0.18	0.03	0.00	0.00
		T	−0.19	0.57	0.61	−0.65	0.96	−2.45	1.18	−1.10	1.22	1.78	0.07	0.61
		*p*	*0.853*	*0.571*	*0.544*	*0.513*	*0.336*	*0.015*	*0.238*	*0.274*	*0.222*	*0.078*	*0.941*	*0.544*
		95% CI	[−0.05, 0.04]	[−0.05, 0.08]	[−0.05, 0.09]	[−0.08, 0.04]	[−0.01, 0.02]	[−0.05, −0.01]	[−0.01, 0.04]	[−0.05, 0.02]	[−0.11, 0.48]	[−0.00, 0.06]	[−0.00, 0.00]	[0.00, 0.00]
**Short** **-** **term memory**	Model 1	β	0.00	0.00	0.03	−0.03	0.00	0.00	0.00					
		T	0.18	0.02	1.23	−1.63	0.71	−0.10	−0.32					
		*p*	*0.860*	*0.987*	*0.227*	*0.111*	*0.479*	*0.923*	*0.750*					
		95% CI	[−0.02, 0.02]	[−0.04, 0.04]	[−0.02, 0.07]	[−0.07, 0.01]	[−0.01, 0.01]	[−0.01, 0.01]	[−0.02, 0.01]					
	Model 2	β	0.00	0.00	0.03	−0.03	0.00	0.00	0.00	−0.02	0.03	0.02		
		T	0.07	0.02	1.21	−1.39	0.80	−0.13	−0.23	−1.79	0.42	1.54		
		*p*	*0.948*	*0.984*	*0.232*	*0.172*	*0.425*	*0.894*	*0.816*	*0.081*	*0.676*	*0.132*		
		95% CI	[−0.02, 0.02]	[−0.04, 0.04]	[−0.02, 0.07]	[−0.07, 0.01]	[−0.01, 0.01]	[−0.01, 0.01]	[−0.02, 0.01]	[−0.04, 0.00]	[−0.11, 0.17]	[−0.01, 0.04]		
	Model 3	β	0.01	0.01	0.03	−0.03	0.00	0.00	0.00	−0.02	−0.02	0.02	0.00	0.00
		T	0.45	0.27	1.27	−1.50	0.86	−0.24	−0.18	−1.86	−0.24	1.53	−0.76	0.00
		*p*	*0.650*	*0.785*	*0.209*	*0.143*	*0.391*	*0.813*	*0.854*	*0.070*	*0.813*	*0.134*	*0.450*	*0.538*
		95% CI	[−0.02, 0.03]	[−0.04, 0.05]	[−0.02, 0.08]	[−0.08, 0.01]	[−0.01, 0.01]	[−0.02, 0.01]	[−0.02, 0.01]	[−0.05, 0.00]	[−0.21, 0.16]	[−0.01, 0.04]	[−0.00, 0.00]	[0.00, 0.00]
**Executive Function**	Model 1	β	−0.02	−0.01	0.00	−0.01	0.01	0.00	0.00					
		T	−1.90	−0.87	0.10	−0.51	1.54	0.00	0.70					
		*p*	*0.059*	*0.383*	*0.921*	*0.613*	*0.124*	*0.999*	*0.483*					
		95% CI	[−0.04, 0.00]	[−0.04, 0.02]	[−0.03, 0.03]	[−0.04, 0.03]	[−0.00, 0.01]	[−0.01, 0.01]	[−0.01, 0.02]					
	Model 2	β	−0.02	−0.01	−0.01	0.00	**0.01**	0.00	0.01	**−0.04**	**−0.22**	0.01		
		T	−1.77	−0.74	−0.49	−0.24	**2.19**	−0.31	1.14	**−5.29**	**−3.83**	1.78		
		*p*	*0.078*	*0.458*	*0.622*	*0.811*	** *0.029* **	*0.756*	*0.253*	** *<0.001* **	** *<0.001* **	*0.078*		
		95% CI	[−0.03, 0.00]	[−0.04, 0.02]	[−0.04, 0.02]	[−0.04, 0.03]	**[0.00, 0.01]**	[−0.01, 0.01]	[−0.00, 0.02]	**[−0.06, −0.03]**	**[−0.33, −0.11]**	[−0.00, 0.03]		
	Model 3	β	−0.02	−0.02	−0.02	−0.01	**0.01**	0.00	0.01	**−0.04**	**−0.19**	0.01	0.00	0.00
		T	−1.94	−1.04	−1.01	−0.37	**2.10**	−0.33	1.31	**−5.26**	**−2.61**	1.75	1.00	−1.07
		*p*	*0.054*	*0.298*	*0.312*	*0.711*	** *0.037* **	*0.741*	*0.191*	** *<0.001* **	** *0.009* **	*0.084*	*0.316*	*0.285*
		95% CI	[−0.04, 0.00]	[−0.05, 0.02]	[−0.05, 0.02]	[−0.04, 0.03]	**[0.00, 0.01]**	[−0.01, 0.01]	[−0.00, 0.02]	**[−0.06, −0.03]**	**[−0.33, −0.05]**	[−0.00, 0.03]	[−0.00, 0.00]	[−0.00, 0.00]
**Processing speed**	Model 1	β	0.00	−0.02	0.02	−0.04	0.00	0.00	0.00					
		T	0.23	−1.02	0.67	−1.55	0.38	0.15	0.43					
		*p*	*0.815*	*0.308*	*0.502*	*0.123*	*0.703*	*0.884*	*0.667*					
		95% CI	[−0.02, 0.03]	[−0.07, 0.02]	[0.00, 0.07]	[−0.08, 0.01]	[−0.01, 0.01]	[−0.01, 0.02]	[−0.01, 0.02]					
	Model 2	β	0.00	−0.02	0.01	−0.03	0.00	0.00	0.00	**−0.03**	−0.13	0.02		
		T	0.33	−0.90	0.41	−1.42	0.80	−0.13	−0.23	**−2.64**	−1.52	1.54		
		*p*	*0.745*	*0.37*	*0.679*	*0.158*	*0.425*	*0.894*	*0.816*	** *0.009* **	*0.129*	*0.132*		
		95% CI	[−0.02, 0.03]	[−0.06, 0.02]	[−0.04, 0.06]	[−0.08, 0.01]	[−0.01, 0.01]	[−0.01, 0.01]	[−0.02, 0.01]	**[−0.06, −0.01]**	[−0.30, 0.04]	[−0.03, 0.04]		
	Model 3	β	0.01	−0.01	0.04	−0.02	0.00	0.00	0.00	**−0.03**	−0.16	0.00	0.00	0.00
		T	0.80	−0.28	1.36	−0.91	0.86	−0.24	−0.18	**−2.56**	−1.45	−0.08	−1.54	0.00
		*p*	*0.425*	*0.783*	*0.173*	*0.364*	*0.391*	*0.813*	*0.854*	** *0.011* **	*0.148*	*0.936*	*0.450*	*0.538*
		95% CI	[−0.02, 0.04]	[−0.05, 0.04]	[−0.02, 0.09]	[−0.07, 0.02]	[−0.01, 0.01]	[−0.02, 0.01]	[−0.02, 0.01]	**[−0.06, −0.01]**	[−0.38, 0.06]	[−0.02, 0.02]	[−0.00, 0.00]	[0.00, 0.00]

## Data Availability

The original contributions presented in the study are included in the article/Appendix A, further inquiries can be directed to the corresponding author.

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
