# Peer review of "The Relationship Between Dietary Patterns, Cognition, and Cardiometabolic Health in Healthy, Older Adults"

_nutrients, 2024, doi:10.3390/nu16223890_

Round 1

Reviewer 1 Report

Comments and Suggestions for Authors

The aim of this research was to evaluate the relationship between dietary patterns and cognition as whether cardiometabolic health markers moderated the relationship in older adults. It was a cross-sectional study of cognitively normal adults and used a validated food frequency questionnaire, cognitive measures, and metabolic health assessment. The dietary patterns were assessed using principal component analysis. A diet that was meat dominant was negatively associated with long-term memory and this relationship was moderated by cardiometabolic risk score. The authors concluded that there is a link among diet, cardiometabolic health and cognitive function in older healthy adults. In general, this is not a particularly novel finding since it has long been known that age-related cognitive function and heart health share risk factors. However, evaluating a dietary pattern relationship offers possible insights to the synergistic effect of dietary components. The authors provide sufficient rationale to conduct this research and the topic and the research adds to this scientific literature to further support the importance of diet and other modifiable factors to healthy aging. The manuscript is well written and organized. The Discussion adequately addressed the strengths and limitations of the study. These are my specific comments:

  1. Given that this research was conducted in a certain geography, how translatable would this be to other populations?

  2. It takes a period of time for diet to affect cognitive function and assessment of usual or long term intakes may be important. Over what period of tie were the participants asked to reflect on their diet?

  3. The environment in which the cognitive testing is performed is important, e.g. time of day, postprandial, noise, temperature. How was the testing standardized among participants?

  4. Is there any reason to believe that there could be a bias for excluding the participants who were removed from the analysis? Were their characteristics different in any way from those who were included?

Author Response

Dear Reviewer 1,

Thank you for your thorough review and insightful questions regarding our manuscript.

Question 1: Generalizability Across Geographies You raised an important point regarding the cultural and geographical generalizability of our findings. We agree that the context in which research is conducted invariably influences its outcomes, particularly in studies involving dietary patterns. In response, we have expanded our discussion to consider the potential cohort-specific issues and the broader implications of cultural diversity. This expanded discussion is now included in the manuscript on page 21, lines 597-604, highlighting the diversity in dietary exposures and the need for further studies across varied geographical and cultural settings.

Question 2: Dietary Assessment Period Regarding the period over which dietary intake was assessed, the Australian Eating Survey (AES) food frequency questionnaire was used to capture participants' dietary consumption over the past six months. This timeframe is clarified in Section 2.3, "Assessment of Diet," on page 4, line 153 of the manuscript, ensuring the assessment reflects long-term dietary patterns rather than short-term fluctuations.

Question 3: Standardization of Cognitive Testing We appreciate your emphasis on the importance of standardizing cognitive testing environments. Our testing conditions were controlled for noise and temperature and conducted by trained research assistants. However, due to multisite data collection logistics, the exact time of day for each session was not controlled, though the sequence of assessments was consistent. These details have been further clarified and can now be found in Section 2.4, lines 178-181, page 4 of the revised manuscript.

Question 4: Exclusion Bias Concerns You also inquired about the potential bias associated with the exclusion of participants from the analysis. We conducted several t-tests to compare demographic and health-related characteristics between included and excluded participants and found no significant differences. These exclusions were strictly due to implausible dietary intake data, affecting only seven participants. While we did not initially include this analysis in the manuscript due to the small number affected, this information has been added to the unpublished supplementary material to ensure transparency.

We hope that the revisions and additions made to the manuscript adequately address your concerns and enhance the robustness and clarity of our findings. Thank you once again for your constructive feedback, which has undoubtedly strengthened our paper.

We look forward to your further comments and hope that our revisions meet your approval for publication.

Reviewer 2 Report

Comments and Suggestions for Authors

The present study examined the associations between dietary habits and cognitive performance among healthy older adults. The text accurately and clearly summarizes the research objectives, methodology, and background, while also including relevant references to support its claims. The paragraphs are well-structured and detail the relationship between cardiometabolic health, diet, and cognitive functions in accordance with the requirements of scientific communication. Its language is precise, well-edited, and meets the expectations for scientific publications. The arguments and results are presented clearly, and the text flows smoothly. It is recommended to add an abbreviation list below the tables to ensure they are interpretable on their own. I recommend this for publication.

Author Response

Dear reviewer, 

Thank you for your thorough review and insightful questions.

Question 1: Generalizability Across Geographies
You raised an important point regarding the cultural and geographical generalizability of our findings. We agree that the context in which research is conducted invariably influences its outcomes, particularly in studies involving dietary patterns. In response, we have expanded our discussion to consider the potential cohort-specific issues and the broader implications of cultural diversity. This expanded discussion is now included in the manuscript on page 21, lines 597-604, highlighting the diversity in dietary exposures and the need for further studies across varied geographical and cultural settings.

Question 2: Dietary Assessment Period Regarding the period over which dietary intake was assessed, the Australian Eating Survey (AES) food frequency questionnaire was used to capture participants' dietary consumption over the past six months. This timeframe is clarified in Section 2.3, "Assessment of Diet," on page 4, line 153 of the manuscript, ensuring the assessment reflects long-term dietary patterns rather than short-term fluctuations.

Question 3: Standardization of Cognitive Testing We appreciate your emphasis on the importance of standardizing cognitive testing environments. Our testing conditions were controlled for noise and temperature and conducted by trained research assistants. However, due to multisite data collection logistics, the exact time of day for each session was not controlled, though the sequence of assessments was consistent. These details have been further clarified and can now be found in Section 2.4, lines 178-181, page 4 of the revised manuscript.

Question 4: Exclusion Bias Concerns You also inquired about the potential bias associated with the exclusion of participants from the analysis. We conducted several t-tests to compare demographic and health-related characteristics between included and excluded participants and found no significant differences. These exclusions were strictly due to implausible dietary intake data, affecting only seven participants. While we did not initially include this analysis in the manuscript due to the small number affected, this information has been added to the unpublished supplementary material to ensure transparency.

We hope that the revisions and additions made to the manuscript adequately address your concerns and enhance the robustness and clarity of our findings. Thank you once again for your constructive feedback, which has undoubtedly strengthened our paper.

We look forward to your further comments and hope that our revisions meet your approval for publication.